# Solution and Parameter Identification of a Fixed-Bed Reactor Model for Catalytic CO$_2$ Methanation Using Physics-Informed Neural Networks

**Son Ich Ngo** [ID] **and Young-Il Lim *** [ID]

Center of Sustainable Process Engineering (CoSPE), Department of Chemical Engineering, Hankyong National University, Jungang-ro 327, Anseong-si 17579, Korea; ngoichson@hknu.ac.kr
* Correspondence: limyi@hknu.ac.kr; Tel.: +82-31-670-5207

**Abstract:** In this study, we develop physics-informed neural networks (PINNs) to solve an isothermal fixed-bed (IFB) model for catalytic CO$_2$ methanation. The PINN includes a feed-forward artificial neural network (FF-ANN) and physics-informed constraints, such as governing equations, boundary conditions, and reaction kinetics. The most effective PINN structure consists of 5–7 hidden layers, 256 neurons per layer, and a hyperbolic tangent (*tanh*) activation function. The forward PINN model solves the plug-flow reactor model of the IFB, whereas the inverse PINN model reveals an unknown effectiveness factor involved in the reaction kinetics. The forward PINN shows excellent extrapolation performance with an accuracy of 88.1% when concentrations outside the training domain are predicted using only one-sixth of the entire domain. The inverse PINN model identifies an unknown effectiveness factor with an error of 0.3%, even for a small number of observation datasets (e.g., 20 sets). These results suggest that forward and inverse PINNs can be used in the solution and system identification of fixed-bed models with chemical reaction kinetics.

**Keywords:** catalytic CO$_2$ methanation; fixed-bed reactor; reaction kinetics; system identification; machine learning; physics-informed neural network



## 1. Introduction

Power-to-gas technology using intermittent surplus renewable electricity has gained attention for mitigating CO$_2$ emissions to the atmosphere [1,2]. The intermittency of renewable energy sources is a major hurdle in their seamless integration with existing energy systems [3]. CO$_2$ methanation [4,5] combining captured CO$_2$ with H$_2$, produced via water electrolysis [6], is an alternative to existing energy systems that could be integrated with renewable electricity sources. CO$_2$ methanation technologies could considerably reduce carbon emissions by encouraging industrial symbiosis from industries with large CO$_2$ footprints [7], such as thermal power plants [8]. Because CH$_4$ is easier to store and transport than H$_2$ [1], the synergistic integration of renewable electricity with a natural gas grid is expected via CO$_2$ methanation [4].

Among the various reactor types, fixed-bed reactors (FBs) are the most commonly used types for CO$_2$ methanation. As the CO$_2$ methanation reaction is thermodynamically favored at low temperatures and high pressures [9], an isothermal fixed-bed reactor (IFB) without a hot spot produces high methane selectivity, exhibits stable operation, and prevents deactivation of catalyst particles through processes such as thermal degradation (i.e., nickel sintering [2]). However, the IFB usually requires high recycling and dilution ratios, and adiabatic reactors to maintain suitable productivity [10–12]. The first commercial CO/CO$_2$ methanation process developed by Lurgi and Sasol (USA, North Dakota) that produces 1.53 billion Nm$^3$/year is composed of an IFB and two adiabatic FB reactors with recycling [10]. In Germany, Linde designed an IFB reactor with an indirect heat exchanger to generate steam from the exothermic heat of reaction [11].

To develop advanced $CO_2$ methanation technologies, researchers have explored the use of modeling and simulations in the optimization of reactor designs. In particular, computational fluid dynamics (CFD) involving process modeling has been used to understand the hydro- and thermo-dynamics of a reactor following geometrical and operational modifications [13]. CFD studies of $CO_2$ methanation have been reported for a single fixed-bed [14], multi-stage fixed-beds [14–17], fixed-beds with a plate-type heat exchanger [18,19], monolith reactors [20,21], gas–solid fluidized beds [1,5,22], three-phase slurry bubble columns [23], and microchannel reactors [24,25]. However, there are several limitations of traditional modeling and CFD, such as (1) high computational cost for three-dimensional (3D) multiscale and multiphase CFD [13,18], (2) difficulties in the efficient discretization of complex geometries [13,21], (3) numerical diffusion and round-off errors stemming from numerical differentiation [26], and (4) difficulties in the identification of physical model parameters [1,17,19,27].

Despite advances in first principles and empirical elucidation, artificial neural network (ANN) models in the category of data-driven models, black-box models, or surrogate models (SMs), have become an alternative functional mapping between input and output data because of their prompt predictions, automated knowledge extraction, and high inference accuracy [28–30]. The structure of ANNs relies on the availability of experimental data, observation data [28], or data generated by first-principle models such as CFD [31,32]. The automatic differentiation (AD) technique, which calculates both functions and their derivative values implementing the chain rule, is used across the ANN layers to efficiently estimate gradients [33]. AD has a lower computational cost than symbolic differentiation and a higher accuracy than numerical differentiation [26].

Recently, ANNs and conservation equations coupled with AD that solve ordinary differential equations (ODEs) and partial differential equations (PDEs) called physics-informed neural networks (PINNs) have been reported [32,34,35]. As PINNs are constrained to respect any symmetries, invariances, or first-principle laws [34], they present great potential for solving chemical engineering problems, which usually deal with complex geometries and physics phenomena. In contrast to common ANNs, PINNs do not depend on empirical data because the initial and boundary conditions are directly used to adjust the network parameters, such as weights and biases [34]. In addition, the extrapolation capability of PINNs is enhanced owing to physical constraints [36]. Nevertheless, there are few applications of PINNs in process modeling and chemical reactor design.

PINNs can be used to solve two types of problems: (1) forward and (2) inverse problems [34]. In the forward problem, the PINN solves ODEs/PDEs, like other numerical solvers of ODEs/PDEs, revealing its inference capability. In the inverse problem, unknown physical model parameters are identified using both the well-trained forward PINN and external input/output datasets. Previous PINN studies focused on a general solution of ODEs/PDEs [32,34,35] and elementary reaction rates [29]. Few researchers have addressed PINNs for complex reaction rate models showing high nonlinearity.

In this study, forward and inverse PINNs coupled with AD were developed for the solution and parameter identification of a highly nonlinear reaction rate model for catalytic $CO_2$ methanation in an IFB reactor. The results obtained from the PINNs were compared with those obtained using a common numerical solver of ODEs (ode15s in MATLAB). The hyper-parameters of the ANN used in the forward PINN, such as (1) the number of hidden layers, (2) number of neurons per hidden layer, (3) activation functions, and (4) number of collocation training points, were systematically determined. In the forward PINN problem, the extrapolation capability was analyzed by narrowing the collocation-training domain and detaching the collocation-training domain from the boundary. In the inverse PINN problem, a reaction effectiveness factor was identified using observation datasets containing 5% and 20% random noises in exact results. The influence of the observation data range on the prediction accuracy of the inverse PINN model was also investigated. This study demonstrates that the forward and inverse PINNs can solve fixed-bed models with highly nonlinear chemical reaction kinetics and identify unknown model parameters.

## 2. Isothermal Fixed-Bed Reactor for CO$_2$ Methanation

A single-tube IFB, which is a type of industrial-scale multi-tubular fixed-bed reactor [37,38], with a length (*L*) of 3 m and a tube diameter ($D_{tube}$) of 0.01 m was filled with spherical catalyst particles with a particle diameter ($d_p$) of 1 mm. The single-tube IFB was assumed to be equipped with a heat exchanger that was able to transfer immediately the heat generated in the exothermic reactions to the coolant. The catalytic CO$_2$ methanation reaction, known as the Sabatier reaction, is [4,14]

$$CO_2 + 4H_2 \rightleftarrows CH_4 + 2H_2O, \ \Delta H_r^{298K} = -165 \text{ kJ·mol}^{-1} \quad (1)$$

Figure 1 shows a tube in the IFB reactor at the reactor and particle scales. The IFB is composed of a single-path tubular reactor at the center and a heat exchanger on the outer wall, where the reactor is externally cooled. The bed voidage (*ε*) between the catalyst particles was assumed to be 0.4 [14]. The operating conditions were set as a temperature (*T*) of 450 °C, a pressure (*P*) of 5 bar, and a volumetric flow rate (*Q*) of 10 Nm$^3$/s. The pure gas reactants were fed to the inlet at a CO$_2$/H$_2$ molar ratio of 1/4.

**Figure 1.** Single tube of an isothermal fixed-bed (IFB) reactor for catalytic CO$_2$ methanation in the reactor and particle scales.

### 2.1. Governing Equations for the Isothermal Fixed-Bed Reactor

The IFB was modeled as a one-dimensional (1D) plug-flow reactor at a steady state [14,38]. The momentum and energy balances were neglected because of the low pressure drop and isothermal conditions, respectively. The mass balances for the *i*th species (*i* = CO$_2$, H$_2$, CH$_4$, and H$_2$O) participating in the CO$_2$ methanation reaction in Equation (1) are formulated as follows:

$$\frac{1}{A_t}\frac{dF_i}{dz} = \eta v_i r \quad (2)$$

where *z* (m) is the reactor tube axial position, $F_i$ (mol/s) is the molar flow rate of a species *i* at position *z*, $A_t$ (m$^2$) is the tube cross-sectional area, $v_i$ is the stoichiometric coefficient of species *i*, and *r* (mol/m$^3$/s) is the volumetric reaction rate. *η* is the effectiveness factor of the chemical reaction, which is defined as the volume-averaged reaction rate with diffusion

within catalyst particles divided by the area-averaged reaction rate at the catalyst particle surface [14]. For the sake of simplicity, the value of $\eta$ was assumed as one in this study.

The boundary conditions for the molar flow rate ($F_i$) of the species at the inlet ($z = 0$) are as follows:

$$F_i|_{z=0} = x_{i,0}F_0 \tag{3}$$

where $x_{i,0}$ and $F_0$ (mol/s) are the inlet mole fraction of gas species $i$ and the total molar flow rate of the inlet gas mixture, respectively. The IFB reactor model expressing the species material balance includes four spatial ODEs in Equation (2) with the boundary condition in Equation (3).

### 2.2. Reaction Kinetics Model

A reaction kinetics model proposed by Koschany et al. (2016) [39] for catalytic $CO_2$ methanation, which was tested within a wide range of Ni contents and industrial operating conditions, was adopted in this study.

$$r = \rho_{cat}(1 - \varepsilon)k \cdot \frac{p_{H_2}^{0.31} p_{CO_2}^{0.16}}{1 + K_{ad} \frac{p_{H_2O}}{p_{H_2}^{0.5}}} \left(1 - \frac{p_{CH_4} p_{H_2O}^2}{p_{H_2}^4 p_{CO_2} K_{eq}}\right) \tag{4}$$

$$k = 6.41 \times 10^{-5} \exp\left(\frac{93.6}{R}\left(\frac{1}{555} - \frac{1}{T}\right)\right) \tag{5}$$

$$K_{ad} = 0.62 \times 10^{-5} \exp\left(\frac{64.3}{R}\left(\frac{1}{555} - \frac{1}{T}\right)\right) \tag{6}$$

$$K_{eq} = 137 \cdot T^{-3.998} \exp\left(\frac{158.7}{RT}\right) \tag{7}$$

where $R$ ($= 8.314 \times 10^{-3}$ kJ/mol/K) is the gas constant, $T$ (K) is the temperature, $p_i$ (bar) is the partial pressure of species $i$, $k$ (mol/$g_{cat}$/s) is the reaction rate constant, $K_{ad}$ (1/$bar^{0.5}$) is the adsorption constant, and $K_{eq}$ is the thermodynamic equilibrium constant. The catalyst density ($\rho_{cat}$) was set to $2300 \times 10^3$ $g_{cat}$/$m^3_{cat}$ [39]. The reaction rate in Equation (4) including inhibition by adsorbed water ($K_{ad}$), equilibrium constant ($K_{eq}$), and non-stoichiometric reaction orders is far from the elementary reaction rate.

## 3. Physics-Informed Neural Networks (PINN) Model

The 1D IFB reactor model coupled with the reaction kinetics in Equations (2)–(7) is typically solved using a stiff ODE solver. In this study, the solution of the 1D IFB reactor model obtained from a stiff ODE solver was compared with that obtained with the PINNs.

The PINN solving the system of ODEs in the IFB reactor model was composed of an FF-ANN, AD, and a governing equation. A strategy for adjusting the hyper-parameters of the FF-ANN was presented in the forward PINN problem. An unknown model parameter (i.e., $\eta$) was identified in the inverse PINN problem. The two PINN structures were the same, while the forward PINN exploited training data self-generated for the initial condition and the inverse PINN used observation data from an external source as the training data.

### 3.1. PINN Structure in the Forward Problem

The architecture of the forward PINN problem is shown in Figure 2. The objective of the forward PINN problem is to solve the given governing equation with initial, boundary,

and operating conditions. The initial conditions were the target values ($F_{i,0}$) over the reactor length ($0 < z \leq L$) except $z = 0$ at the beginning of the reaction, which are given as follows:

$$F_{i,0}(z) = \begin{cases} 94.74 \frac{mol}{s} \text{ for } CO_2 \\ 378.9 \frac{mol}{s} \text{ for } H_2 \\ 0 \text{ for } CH_4 \\ 0 \text{ for } H_2O \end{cases}, 0 < z \leq L \tag{8}$$

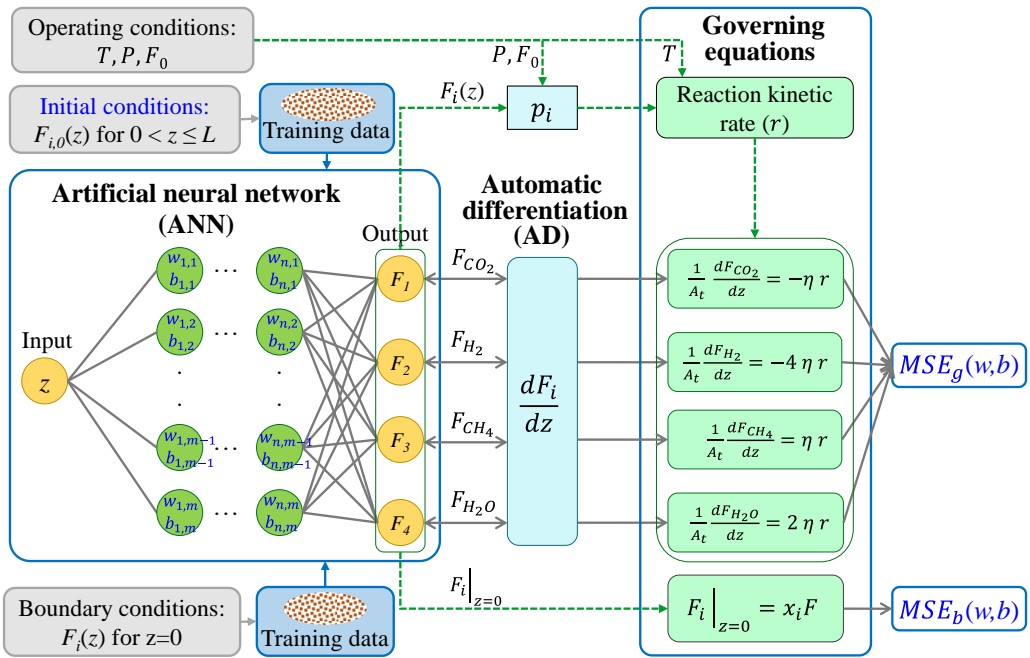

**Figure 2.** Architecture of the physics-informed neural network (PINN) forward problem for $CO_2$ methanation in an isothermal fixed-bed (IFB) reactor.

However, any initial condition must be acceptable in theory to adjust $w$ and $b$ because the forward PINN converges to a solution ($F_{i,PINN}$) satisfying the physics-informed constraints. The Dirichlet boundary conditions at $z = 0$ were implemented separately in the PINN:

$$F_i(0) = x_{i,0}F_0 = \begin{cases} 94.74 \frac{mol}{s} \text{ for } CO_2 \\ 378.9 \frac{mol}{s} \text{ for } H_2 \\ 0 \text{ for } CH_4 \\ 0 \text{ for } H_2O \end{cases} \tag{9}$$

The operating conditions of the IFB, such as $T$, $P$, and $F_0$, were used to calculate the partial pressure ($p_i$) and reaction kinetic rate ($r$). The FF-ANN structure contained one input ($z$), four outputs ($F_i$), $n$ hidden layers, and $m$ neurons for each layer. The input and output datasets of the FF-ANN were randomly sampled from the initial and boundary conditions, Equations (8) and (9), respectively, in the training stage. The activation function ($f_a$), such as *the sigmoid* and hyperbolic tangent (*tanh*), was applied for each neuron. The weights ($w_{j,k}$) and biases ($b_{j,k}$) for the $j$th hidden layer and the $k$th neuron must be adjusted to minimize the loss function (*Loss*). The AD for spatial derivatives ($\frac{dF_i}{dz}$) was calculated via the reverse accumulation mode, which propagates derivatives backward from a given output [26]. The governing equations as the physics-informed part of the ANN included the reaction kinetic rate ($r$) in Equation (4), the four ODEs in Equation (2), and the boundary conditions in Equation (3).

The optimized weights and biases ($w^*$ and $b^*$) were obtained from the following optimization problem:

$$\{w^*, b^*\} = \underset{w,b}{\text{argmin}}\{Loss = MSE_g(w,b) + MSE_b(w,b)\} \tag{10}$$

$$MSE_g(w,b) = \frac{1}{N_{train}} \sum_{j=1}^{N_{train}} \sum_{i=1}^{N_{comp}} \left| \frac{1}{A_t}\left(\frac{dF_i}{dz}\right)_j - \eta\nu_i r_j \right|^2 \tag{11}$$

$$MSE_b(w,b) = \frac{1}{N_{bnd}} \sum_{k=1}^{N_{bnd}} \sum_{i=1}^{N_{comp}} \left| F_{i,k}|_{z=0} - x_{i,0} F_0 \right|^2 \tag{12}$$

where $MSE_g$ and $MSE_b$ are the mean squared errors for the governing equation and boundary condition, respectively. $N_{train}$, $N_{comp}$, and $N_{bnd}$ are the number of training data sets, species (or components), and boundary condition sampling points, respectively. The loss function (*Loss*) sums $MSE_g$ and $MSE_b$.

In Equation (8), for the initial condition, 1000–10,000 training data were randomly and uniformly sampled for the adjustment of the ANN parameters ($w$ and $b$) and determination of the hyper-parameters ($n$, $m$, $f_a$, and $N_{train}$). An Adam optimizer [40] was used to solve Equation (10), which combines a stochastic gradient descent with adaptive momentum, because of its good convergence speed [41], as confirmed in several PINN models [29,35,42,43]. An initial learning rate of 0.001 and decay rate of 0.005 were chosen for the Adam optimizer.

Negative intermediate outputs ($F_i$) appeared frequently when the stochastic gradient optimizer was used in the PINN. However, a negative $F_i$ was unfavorable for solving the IFB reaction model with the reaction kinetics. In addition, as the ODE system of the reactor model with chemical reaction rates was stiff, it was desirable to avoid negative $F_i$ and improve the convergence of the PINN. An exponential mapping of the output values from each hidden layer [29] was used:

$$a_{j,l} = \exp\left( f_a \sum_{k=1}^{m} \left[ w_{j,k} a_{j-1,k} + b_{j,k} \right] \right) \tag{13}$$

where $a_{j,l}$ is the value exiting the *l*th neuron of the *j*th hidden layer.

Gusmao et al. (2020) [29] presented forward and inverse PINNs to create an SM for the solution of chemical reaction kinetics and to acquire kinetic parameters from experimental data. In our study, the PINN concept was applied to a complex and stiff reaction kinetic problem for $CO_2$ methanation.

### 3.2. PINN Structure in the Inverse Problem

The weights ($w$) and biases ($b$) of the FF-ANN are the optimized variables in the forward PINN problem, whereas unknown model parameters are identified in the inverse PINN using the optimized weights ($w^*$) and biases ($b^*$) obtained from the forward PINN. The inverse PINN included the governing equations, boundary, and operating conditions that were used in the forward PINN, as shown in Figure 3. Rather than using the initial condition as the training data, the inverse PINN problem uses observation data from an external source, such as experimental data. Therefore, the values of $z$ and $F_i(z)$ are different from those in Figure 2.

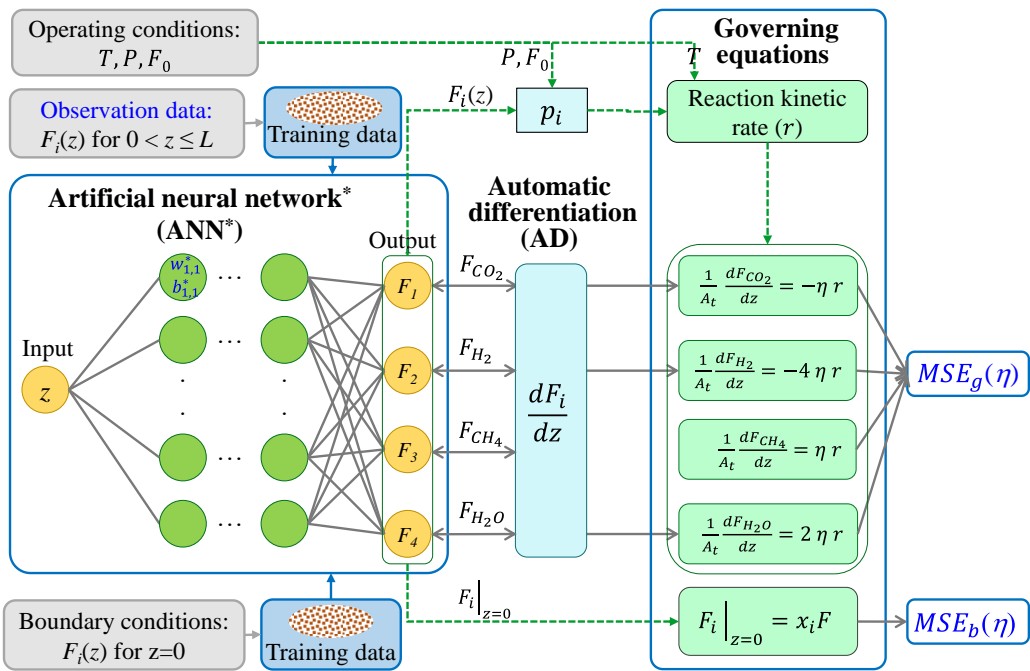

**Figure 3.** Architecture of the physics-informed neural networks (PINN) inverse problem for $CO_2$ methanation in an isothermal fixed-bed (IFB) reactor.

In the inverse PINN, the effectiveness factor ($\eta$) as an unknown model parameter was identified using the following optimization with a loss function:

$$\{\eta^*\} = \underset{\eta}{\arg\min}\{Loss = MSE_g(\eta) + MSE_b(\eta)\} \tag{14}$$

$$MSE_g(\eta) = \frac{1}{N_{obs}} \sum_{j=1}^{N_{obs}} \sum_{i=1}^{N_{comp}} \left| \frac{1}{A_t}\left(\frac{dF_i}{dz}\right)_j - \eta \nu_i r_j \right|^2 \tag{15}$$

$$MSE_b(\eta) = \frac{1}{N_{bnd}} \sum_{k=1}^{N_{bnd}} \sum_{i=1}^{N_{comp}} \left| F_{i,k} \big|_{z=0} - x_{i,0} F_0 \right|^2 \tag{16}$$

where $N_{obs}$ is the number of observation data points (or the experimental data). $MSE_g$ was evaluated for the observation data. The $MSE_b$ in the inverse PINN used the same training data for the Dirichlet boundary condition that was used in the forward problem. The loss function (*Loss*) sums $MSE_g$ and $MSE_b$ as functions of $\eta$.

### 3.3. Hyper-Parameters Setting and Accuracy of PINN Solution

For the FF-ANN, a mini-batch size of 128, which had a minor effect on the PINN training results, was used. The number of training epochs was set to 1000. The number of the hidden layers (*n*) ranged from 2 to 11. The number of neurons (*m*) for each layer was 64–256. The *sigmoid* and *tanh* functions were considered the activation functions ($f_a$). The number of training data points ($N_{train}$) varied from 1000 to 10,000. The training data were used to adjust *w* and *b*, and to determine the hyper-parameters ($m$, $n$, $f_a$, and $N_{train}$), because the validation data for determining hyper-parameters were not necessary in the PINN, which provides a solution for physical models.

In the FF-ANN, the biases (*b*) were initialized to zeros and the weights (*w*) were initialized by the following commonly used heuristic called Xavier's method [44]:

$$w = U\left[ -\sqrt{\frac{6}{N_{in} + N_{out}}}, \sqrt{\frac{6}{N_{in} + N_{out}}} \right] \tag{17}$$

where $U$ is the uniform distribution in the interval of $\pm\sqrt{\frac{6}{N_{in}+N_{out}}}$. $N_{in}$ and $N_{out}$ are the neuron numbers of the previous and present layers, respectively. A pseudo-random generator "*philox*" [45] with 10 rounds and a seed value of "1234" was used to provide the same initial weights for all trainings according to different hyper-parameters. The training data were sampled using Sobol's quasi-random sequence generator [46], which filled the $z$ space in a highly uniform manner.

The PINNs were implemented using the deep learning toolbox of MATLAB (Mathworks, R2021a, Natick, MA, USA, 2021). The PINNs were executed on a single NVIDIA Quadro RTX 6000 GPU device. The computational time was 5 min to 3 h to train each forward PINN according to the number of hidden layers and neurons. The inverse PINN model required less than 20 s.

The governing equation in Equation (2) with the boundary conditions in Equation (3) was also solved using a stiff ODE numerical solver, ode15s in MATLAB, with a strict relative and absolute tolerance of $1 \times 10^{-8}$. The solution was considered as an exact solution. The accuracy of the PINN model was measured using an $L_2$ relative error norm ($L_{2,rel}$) [30] between the PINN solution ($F_{i,PINN}$) and stiff ODE solution ($F_{i,ODE}$):

$$L_{2,rel} = \sqrt{\frac{\sum_{i=1}^{N_{comp}} \sum_{j=1}^{N_{test}} \left(F_{i,PINN}^j - F_{i,ODE}^j\right)^2}{\sum_{i=1}^{N_{comp}} \sum_{j=1}^{N_{test}} \left(F_{i,ODE}^j\right)^2}} \tag{18}$$

$N_{test}$ (=1000) is the number of test data in the forward PINN, whereas $N_{test}$ is the number of observation data points ($N_{obs}$) in the inverse PINN. The test and observation data were generated uniformly over a given range of reactor lengths ($z$).

## 4. Results and Discussion

The hyper-parameters ($m$, $n$, $f_a$, and $N_{train}$) of the forward PINN were first determined. Then, the extrapolation performance of the forward PINN was investigated using 10,000 training data points sampled at a limited reactor length ($z$) and 1000 test data points sampled at the full reactor length ($0 \leq z \leq 3$). Using the optimized structure of the forward PINN, the effects of the number and range of the observation data on unknown model parameters (e.g., $\eta$) were examined in the inverse PINN problem.

### 4.1. Determination of Hyper-Parameters

Activation functions such as the *sigmoid* and *tanh* functions were first tested in an FF-ANN structure with four hidden layers and 64 neurons per layer. Then the number of layers and neurons were determined for the FF-ANN using an activation function selected previously. The *sigmoid* activation function showed good performance for a specific FF-ANN structure (e.g., four hidden layers and 64 neurons), while the *tanh* activation function exhibited good results for almost all network structures.

Figure 4 shows the comparison between *sigmoid* and *tanh* in terms of (a) the loss function and (b) $F_i$ along the reactor length ($z$) for 1000 test data points evenly distributed in $0 \leq z \leq 3$. The 10,000 training data points used to adjust $w$ and $b$ were distributed by Sobol's quasi-random sequence generator, as mentioned earlier. The loss function (*Loss*) obtained using *tanh* was lower by three orders of magnitude than that obtained using *the sigmoid* function (Figure 4a). Here, the iteration number is the number of mini-batches multiplied by the number of epochs. As a result, the mole flow rates ($F_{i,PINN}$) obtained from the PINN with *tanh* were closer to the ODE solution ($F_{i,ODE}$) than those obtained from the PINN with the *sigmoid* function (Figure 4b). The $L_{2,rel}$ of the *sigmoid* and *tanh* functions were 0.05251 and 0.02372, respectively. Therefore, the *tanh* activation function for all the hidden layers was chosen for further investigation. The mixed activation functions and other activation functions, such as the rectified linear unit (ReLU), were outside the scope of this study.

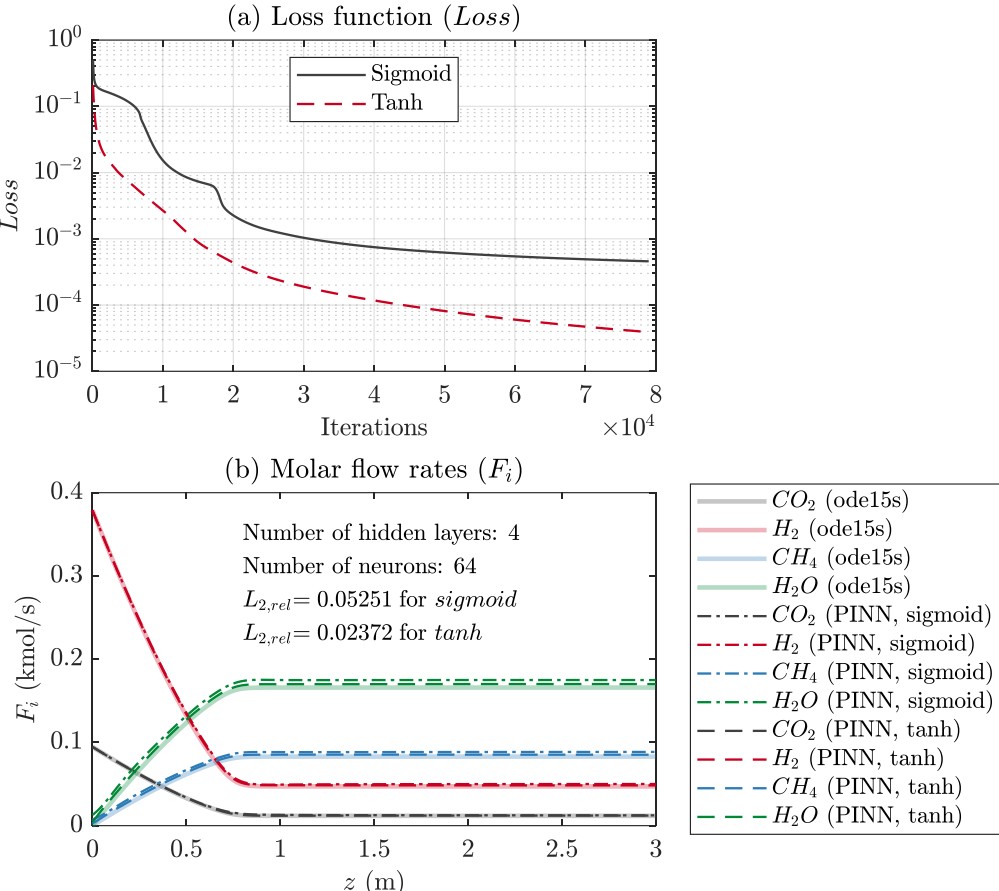

**Figure 4.** (**a**) History of the loss function (*Loss*) and (**b**) comparison of the mole flow rates ($F_i$) of the exact ODE solutions ($F_{i,ODE}$) and PINN solutions ($F_{i,PINN}$) with the *sigmoid* and *tanh* activation functions.

Figure 5 shows the influence of the number of hidden layers and neurons in each layer on the loss function (*Loss*), $L_2$ relative error ($L_{2,rel}$), and training time ($t$). The loss function and training time were obtained from the PINN with *tanh* for 10,000 training data points. $L_{2,rel}$ was measured for 1000 test data compared to the ODE solution. The red spot indicates the forward PINN structure with $n = 7$ layers and $m = 256$ neurons achieved a minimum loss.

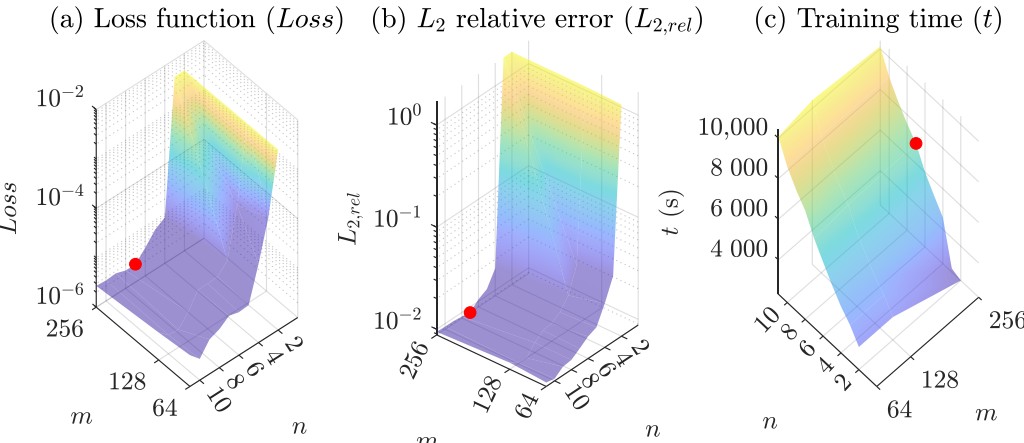

**Figure 5.** (**a**) Loss function (*Loss*), (**b**) $L_2$ relative error ($L_{2,rel}$), and (**c**) training time ($t$) with respect to the number of layers ($n$) and neurons ($m$) for each layer.

*Loss* and $L_{2,rel}$ sharply decrease for $2 \leq n \leq 5$ and slowly converge to a certain value for all the investigated numbers of neurons (see Figure 5a,b). The number of neurons (*m*) weakly influenced *Loss* and $L_{2,rel}$, which were the lowest for the forward PINN structure with 256 neurons. Although the minimum values of $Loss = 2.3 \times 10^{-6}$ and $L_{2,rel} = 0.00881$ were obtained with the forward PINN structure with seven layers and 256 neurons, the variation in $L_{2,rel}$ was negligible for forward PINN structures with more than four layers, at less than 0.69%. The computational time (*t*) for the training increased almost linearly with the number of hidden layers (*m*). However, the training time (*t*) did not increase as the number of neurons in each layer increased. This may be attributed to the fast convergence achieved with a high number of neurons. The FF-ANN structure with seven hidden layers and 256 neurons was chosen for the $CO_2$ methanation IFB reactor model.

Figure 6 compares the performance of the PINN with two, five, and seven hidden layers, using 256 neurons for each layer. The $L_{2,rel}$ decreases with an increase in the number of hidden layers, as shown in Figure 5. The solution obtained from the PINN with two layers and 256 neurons deviated significantly from the ODE solution (Figure 6a). The solution obtained from the PINN with seven layers showed excellent agreement with the ODE solution at high computational cost (Figure 6c).

Using the PINN with seven layers and 256 neurons, the influences of the number of training data points ($N_{train}$) on the loss function (*Loss*) and training time (*t*) are depicted in Figure 7. As $N_{train}$ increases, the *Loss* converges, and *t* increases proportionally. For the error between the PINN and ODE solutions to be sufficiently small, it is desirable that $N_{train}$ for the 1D reactor model be over 5000.

### 4.2. Computational Efficiency of the PINN

A single training time (*t*) was approximately two hours for the PINN with *m* = 256, *n* = 7, *tanh*, and $N_{train}$ = 10,000, as shown in Figure 7. A substantial computational time was required to determine all hyper-parameters of PINN, whereas the ODE numerical solver (e.g., ode15s) showed fast calculation due to no training stage. However, once the hyper-parameters were determined, and weights (*w*) and biases (*b*) were optimized, the PINN surrogate model against the ODE numerical solver had the advantage in computational time.

In Figure 8, the computational times of the PINN surrogate model and the ODE numerical solver are compared with respect to the number of spatial points from 1000 to 200,000 with an interval of 1000. The PINN with *n* = 7, *m* = 256 and *tanh* activation function was used. The calculation time (*t*) increases with the increase in the number of spatial points or $N_{test}$. The calculation time of PINN is less than 0.1 s at $N_{test}$ = 200,000, while that of ODE solver with 200,000 spatial points is approximately 0.95 s. The PINN surrogate model has computational efficiency over the ODE solver and it will be useful for optimization problems repeating the calculation of ODEs/PDEs.

### 4.3. Extrapolation Capability of the PINN

If a sufficient amount of training data is provided (see Figure 7), the forward PINN guarantees a solution ($F_{i,PINN}$) that satisfies the governing equation, as mentioned earlier. The PINN has an extrapolation capability when applied for the range out of training data, which is similar to solving first-principle laws in a computational domain.

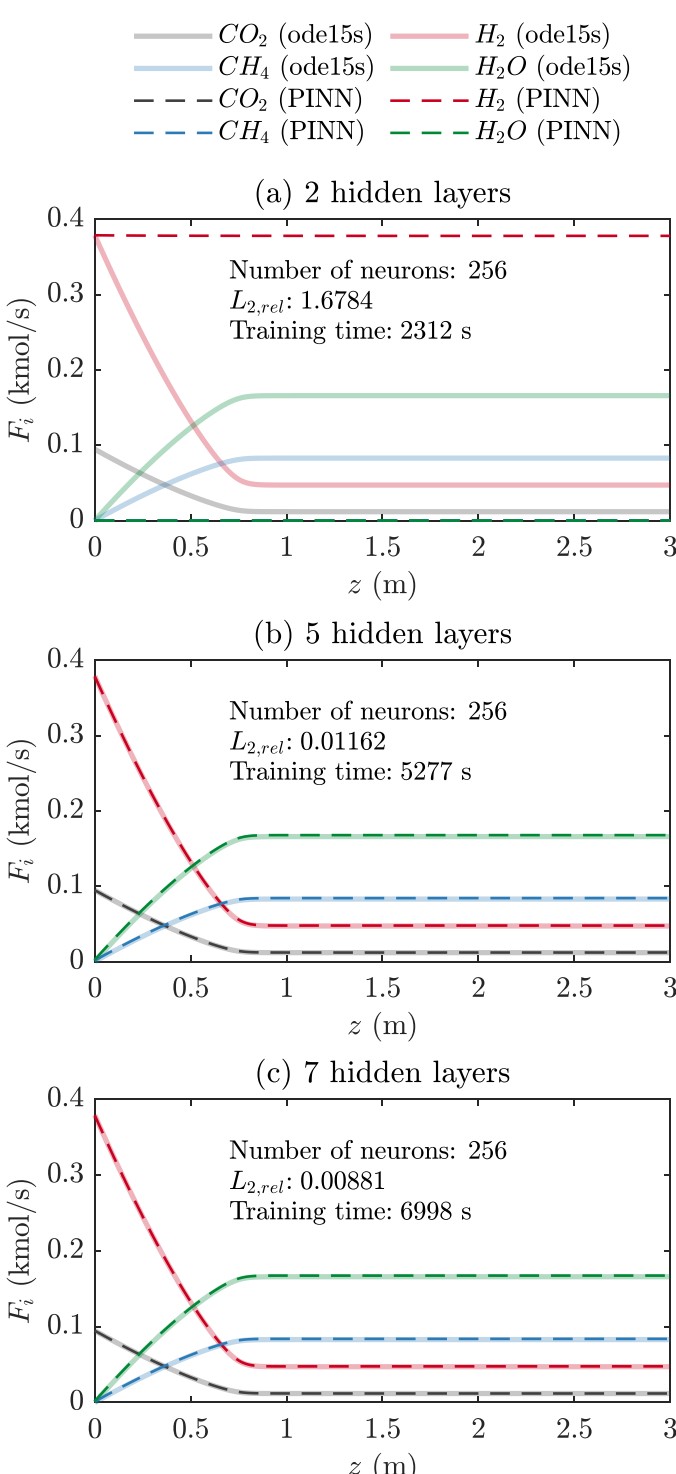

**Figure 6.** Performance of PINN models with different numbers of hidden layers: (**a**) 2 hidden layers, (**b**) 5 hidden layers, and (**c**) 7 hidden layers.

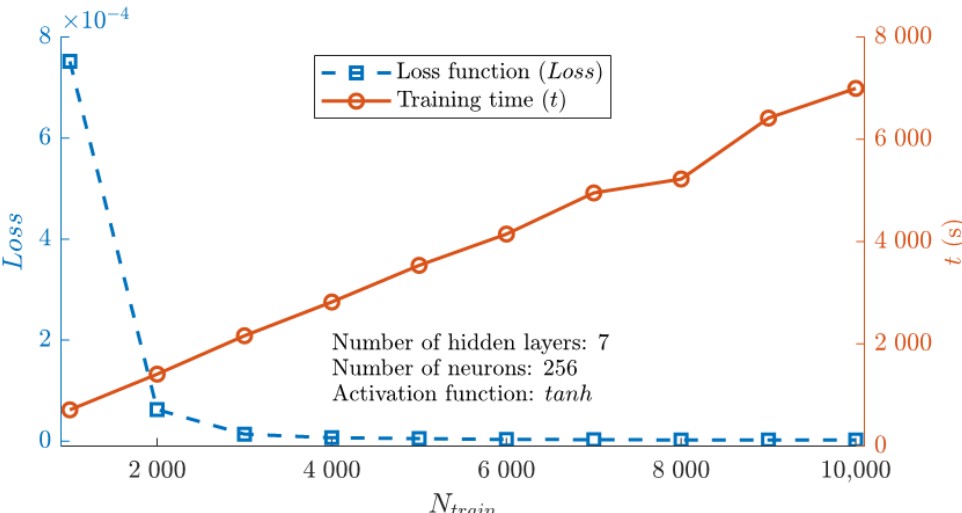

**Figure 7.** Influence of the number of training data points ($N_{train}$) on the loss function (*Loss*) and training time ($t$).

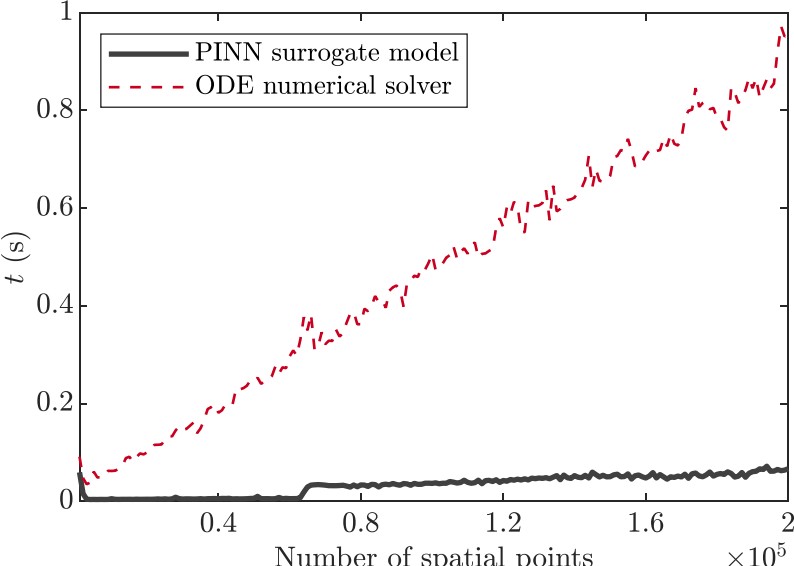

**Figure 8.** Comparison of calculation time between PINN surrogate model and ODE numerical solver with respect to the number of spatial points.

Figure 9 shows the performance of the forward PINN for 10,000 training data points in a limited range of $z$ and 1000 test data points in a full range of $z$ ($0 \le z \le 2$), using five hidden layers, 256 neurons per layer, and the *tanh* activation function. The collocation range of the training data starts from $z = 0$ and ends at $z = 0.5$–1.0, with an interval of 0.1 in Figure 9a–f. Even though the PINN was trained within one sixth ($0 \le z \le 0.5$) of the full range, the PINN output ($F_{i,PINN}$) for the test data of the full range ($0 \le z \le 2$) agrees well the ODE solution ($F_{i,ODE}$) outside the training range (Figure 9a). $L_{2,rel}$ decreases, and the training time tends to increase as the training range increases.

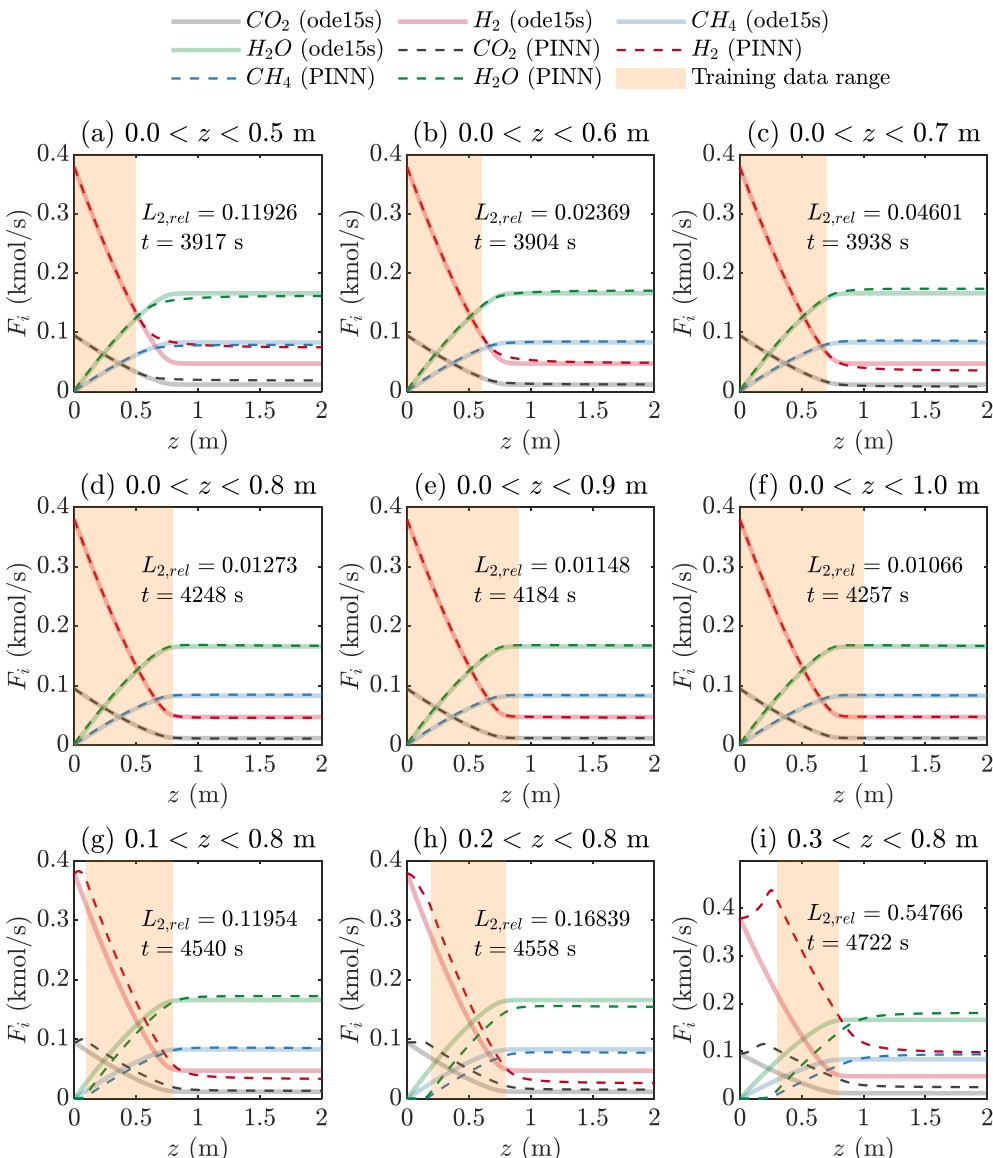

**Figure 9.** Performance of the forward PINN model using five hidden layers, 256 neurons, and the *tanh* activation function for 10,000 training data points in a limited range of the reactor length ($z$) and 1000 test data points in the full range of $z$: (**a**) $0 < z < 0.5$ m, (**b**) $0 < z < 0.6$ m, (**c**) $0 < z < 0.7$ m, (**d**) $0 < z < 0.8$ m, (**e**) $0 < z < 0.9$ m, (**f**) $0 < z < 1.0$ m, (**g**) $0.1 < z < 0.8$ m, (**h**) $0.2 < z < 0.8$ m, and (**i**) $0.3 < z < 0.8$ m.

Figure 9g–i show the performance of the PINN that was trained for three data ranges detached from $z = 0$, reducing the width of the range of $z$. The value of $F_i|_{z=0}$ is correct for all data ranges detached from $z = 0$ owing to the boundary condition in Equation (3). However, $F_{i,PINN}$ between $z = 0$ and the beginning point of the training data is far from $F_{i,ODE}$, and the errors between $F_{i,PINN}$ and $F_{i,ODE}$ are persistent in the other ranges. As the governing equation used in this study is a type of initial value problem, sufficient training data close to $z = 0$ must be provided to obtain reliable PINN solutions.

The extrapolation capability of the PINN is remarkable, unlike that of common ANNs [28,30]. The accuracy of the PINN solution is closely related to the range and distribution of the training data [35]. The present forward PINN model for solving governing equations can overcome the drawbacks of traditional CFD modeling and simulation methods, as mentioned in the introduction. Once the PINN parameters (hyper-parameters, weights, and biases) are optimized, the PINN instantly predicts outputs ($F_i$) corresponding

to inputs ($z$), which can be used as an excellent SM of the governing equation. Because training data at spatial collocation points ($z$) are generated independently against the specific spatial domain, the forward PINN model is appropriate for solving governing equations with complex geometries or moving boundary conditions [47]. In addition, numerical diffusion and round-off errors are minimized in PINNs with the aid of AD [48].

### 4.4. Identification of Unknown Model Parameters with the Inverse PINN

The trained PINN with seven layers, 256 neurons per layer, and the *tanh* activation function was used in the inverse PINN problem to identify an unknown parameter (i.e., $\eta$). The number of epochs, initial learning rate, and decay rate of the inverse PINN model were set to 500, 0.03, and 0.005, respectively. The initial $\eta$ value was assumed to be $1 \times 10^{-6}$.

Figure 10 shows the influence of the number of observation data points ($N_{obs}$) and noise ratios on the identification performance of the effectiveness factor ($\eta$). $N_{obs}$ had a range of 20–1000. Initially, the ODE solution ($F_{i,ODE}$) for the governing equation with $\eta = 1$ and $0 \leq z \leq 2$ was obtained using a stiff ODE solver (i.e., ode15s in MATLAB), which is the mean value of $F_i(z)$. The observation data were randomly and uniformly generated, satisfying a normal distribution with a mean of $F_i(z)$ and standard deviation as the noise ratios of 5% and 20%. The $F_{i,PINN}$ displayed by the solid, dashed, and dotted lines in Figure 10 was acquired from the forward PINN with $\eta^*$ obtained from the inverse PINN.

It is expected that the six inverse PINN problems in Figure 10 result in $\eta^* = 1$ because all observation data were generated for $\eta = 1$. $N_{obs}$ influence the value of $\eta^*$ more significantly than the noise ratio. If the number of observation data is the same, the noise ratio hardly affects $\eta^*$. Even though a small amount of observation data ($N_{obs} = 20$) were used, the error between the exact and predicted $\eta$ values was only 0.3% (see Figure 10a,b). The inverse PINN model inherits the accuracy of the forward PINN model. Thus, the inverse PINN model with well-trained weights and biases can accurately identify model parameters, even for a small amount of observation data.

The effect of the collocation range of observation data on $\eta^*$ was investigated for the inverse PINN model, as shown in Figure 11. The collocation range of the observation data significantly influences the identification of $\eta$. High accuracy was achieved when the collocation range of the observation data was close to the boundary ($z = 0$), as shown in Figure 11a,b. When observation data far from the boundary are provided, it was difficult for the inverse PINN model to identify the model parameter ($\eta$), as shown in Figure 11c. The inverse PINN problem took approximately 20 s for 1000 observations. If the forward PINN is well-trained and the observation data are properly provided, the inverse PINN model can identify unknown model parameters more efficiently than other computationally intensive methods such as CFD.

### 4.5. Extension of PINN to Different Effectiveness Factors

The current PINN can be extended to the ODE solutions for different process conditions, such as the effectiveness factor ($\eta$). One more neuron for $0 \leq \eta \leq 1$ was added into the input layer, which results in the input layer with $z$ and $\eta$, and the identical output layer ($F_i$). The boundary condition with the combination of $z$ and $\eta$ was used as follows:

$$F_i(0, \eta_0) = x_{i,0}F_0 \text{ and } F_i(0, \eta_1) = x_{i,0}F_0 \tag{19}$$

where $\eta_0 = 0$ and $\eta_1 = 1$ are the bounds of $\eta$.

The previous optimized network structure with five hidden layers, 256 neurons per layer, and *tanh* activation function was used. The number of training points ($N_{train}$) was increased from 10,000 to 30,000 because the two input variables ($z$ and $\eta$) were applied. The number of epochs was also increased from 1000 to 2000 to enhance the convergence during training. Figure 12 shows the PINN predictions $F_i(z)$ for $\eta = 0.6$, 0.8, and 1.2, where $N_{test} = 1000$. The PINN captured the $F_i(z)$ at any $\eta$. Excellent prediction for $F_i(z)$ at $\eta = 1.2$, as an extrapolation was observed, as shown in Figure 12c. The PINN surrogate

model can be used for an optimization problem of the $CO_2$ methanation process with computational efficiency.

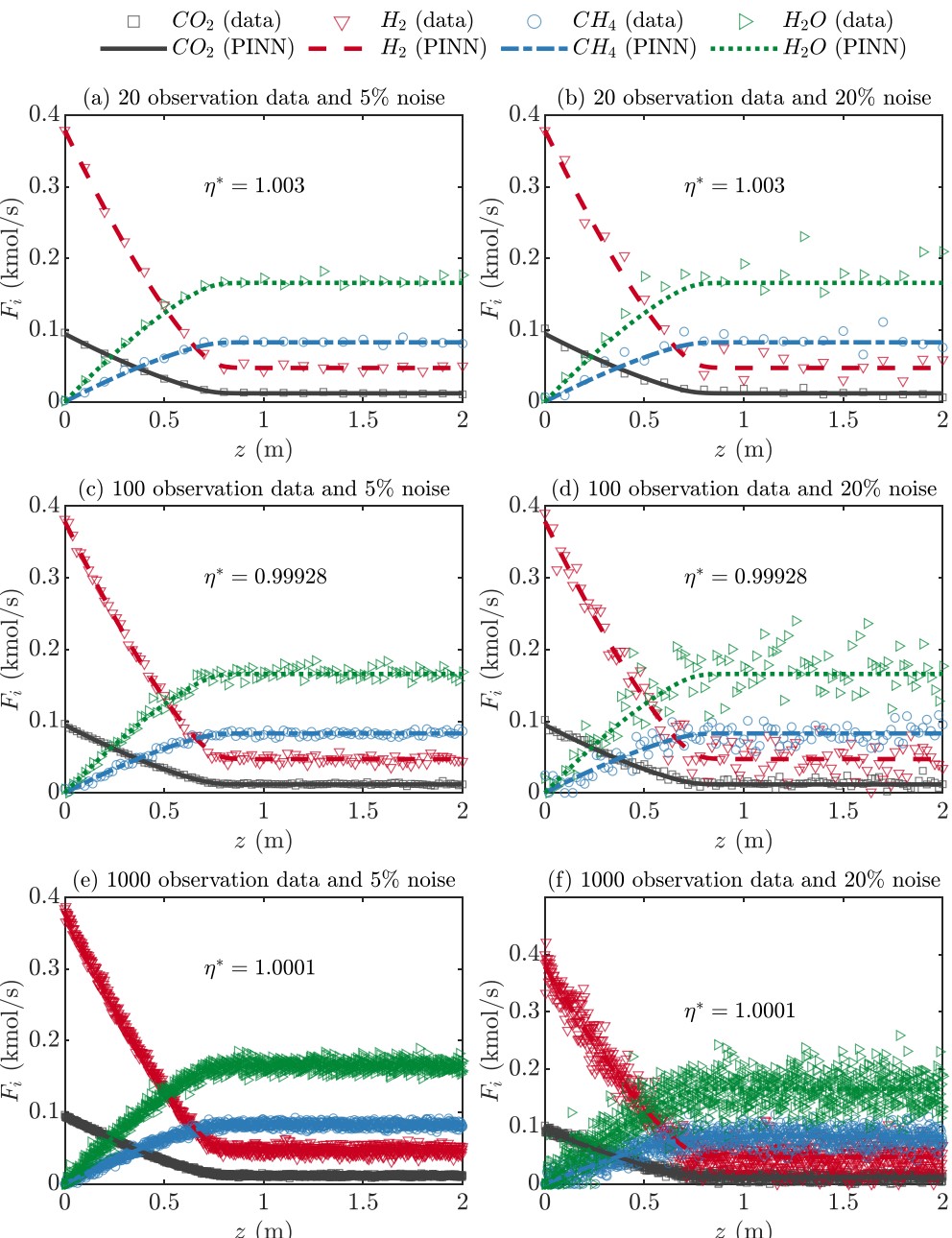

**Figure 10.** Influence of the number of observation data points ($N_{obs}$) and noise ratios on the identification performance of the effectiveness factor ($\eta$) of an inverse PINN with seven layers, 256 neurons per layer, and the *tanh* activation function: (**a**) 20 data and 5% noise, (**b**) 20 data and 20% noise, (**c**) 100 data and 5% noise, (**d**) 100 data and 20% noise, (**e**) 1000 data and 5% noise, and (**f**) 1000 data and 20% noise.

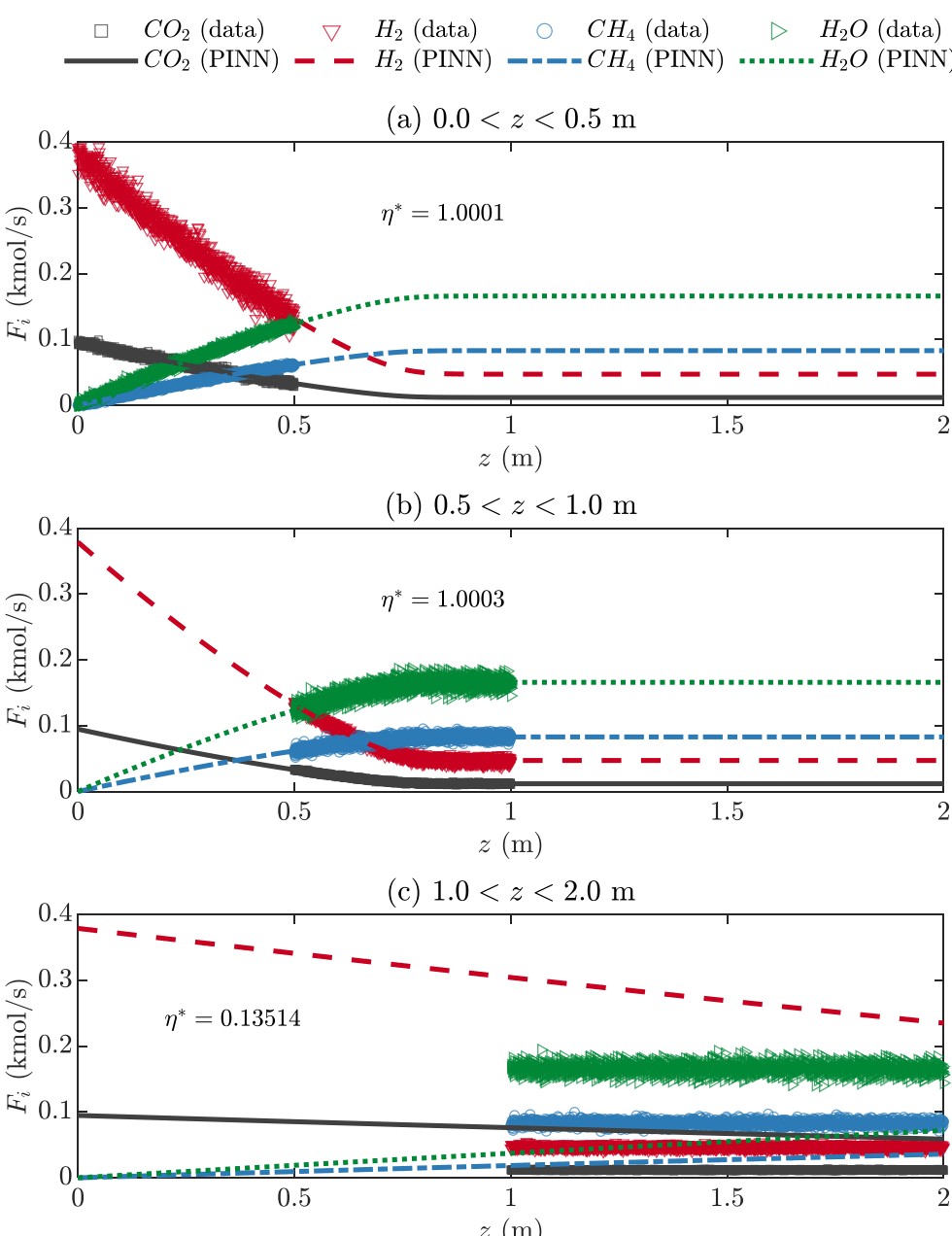

**Figure 11.** Performance of the inverse PINN model with seven hidden layers, 256 neurons, and the *tanh* activation function for 1000 observation data points in a limited range of the reactor length ($z$): (**a**) $0 < z < 0.5$ m, (**b**) $0.5 < z < 1.0$ m, and (**c**) $1.0 < z < 2.0$ m.

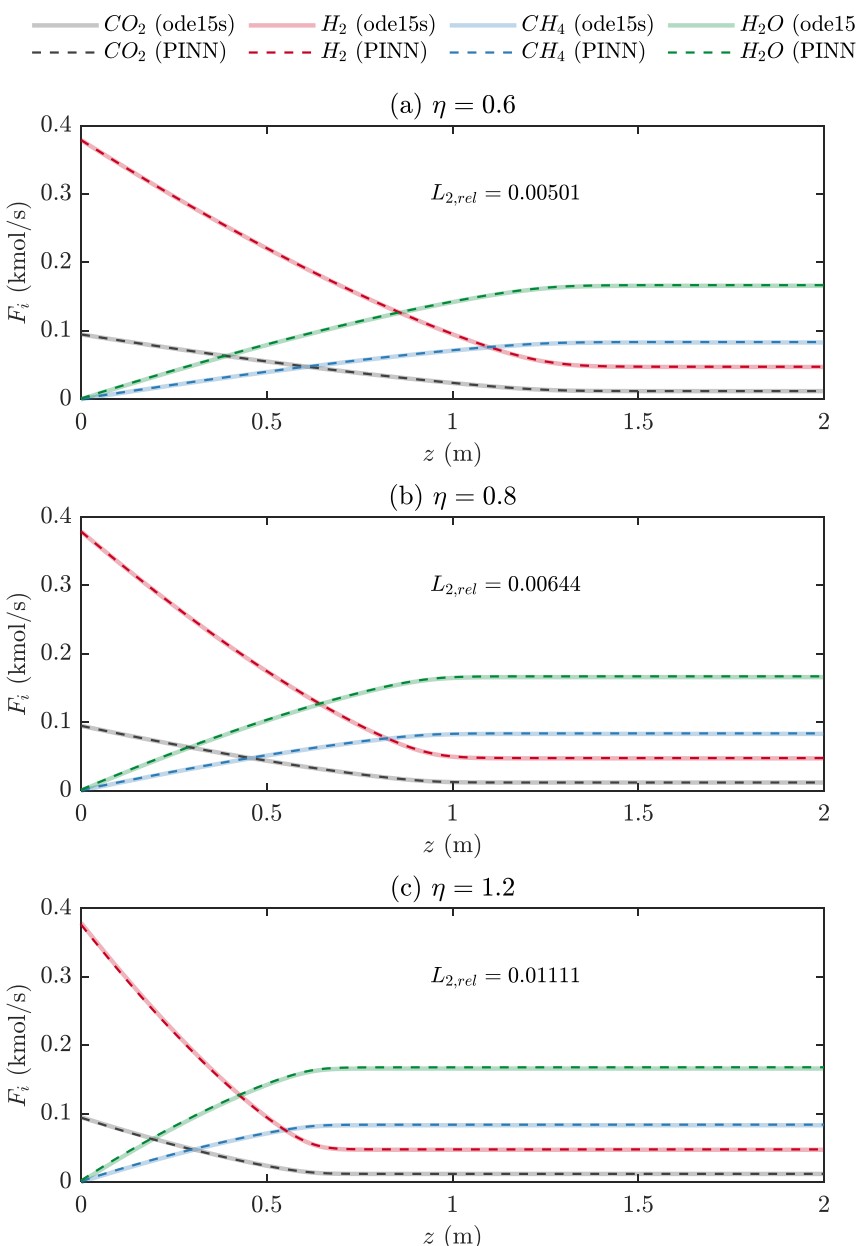

**Figure 12.** Performance of forward PINN models with two inputs ($z$ and $\eta$) at $\eta = 0.6$ (**a**), 0.8 (**b**), and 1.2 (**c**).

## 5. Summary and Conclusions

A physics-informed neural network (PINN) was developed for an isothermal fixed-bed (IFB) reactor model for catalytic $CO_2$ methanation. The PINN was composed of a feed-forward artificial neural network (FF-ANN), automatic differentiation (AD) for derivatives, and governing equations with a stiff reaction kinetic rate. The loss function of the PINN included two mean squared errors (MSEs) for the governing equations and boundary conditions. The one-dimensional reactor was initialized at a molar flow rate that was the same as the boundary condition at the reactor inlet.

For the forward problem, the PINN solved the material balance expressed by ordinary differential equations (ODEs) for the IFB reactor model, where hyper-parameters, weights, and biases of FF-ANN were determined. For the inverse problem, the PINN used the weights and biases of the trained forward PINN model and identified unknown model parameters, such as the effectiveness factor of the $CO_2$ methanation reaction, using observation data. Future work is to implement the PINN for the solution of the fixed-bed reactor

model with a wide range of operating conditions (temperature, pressure, flow rate, and inlet composition), where the reactor model includes the heat and mass balances. The key conclusions drawn are as follows.

- The PINN with the *tanh* activation function, 5–7 hidden layers, and 256 neurons per hidden layer was found to have the most effective combination of hyper-parameters for the IFB reactor model.
- The reliability of the PINN depended on the number and range of the training data.
- When the molar flow rates of the reactor were predicted as out of the range of training data, the forward PINN model exhibited an excellent extrapolation performance because the PINN provides a solution satisfying physics-informed constraints.
- The inverse PINN model identified unknown model parameters when the observation data were properly provided to the inverse PINN.
- The training time of the forward PINN was almost proportional to the number of hidden layers and the number of training data points. The training time of the inverse PINN was relatively short, and the inverse PINN was more efficient at identifying unknown model parameters compared to other numerical methods such as computational fluid dynamics (CFD).
- The present PINN model was extended to different process conditions such as effectiveness factors.
- The current approach is useful for building a surrogate model for $CO_2$ methanation process design and optimization.

**Author Contributions:** Conceptualization, S.I.N. and Y.-I.L.; methodology, S.I.N. and Y.-I.L.; software, S.I.N.; validation, S.I.N. and Y.-I.L.; formal analysis, S.I.N.; investigation, S.I.N. and Y.-I.L.; resources, S.I.N. and Y.-I.L.; data curation, S.I.N.; writing—original draft preparation, S.I.N.; writing—review and editing, Y.-I.L.; visualization, S.I.N. and Y.-I.L.; supervision, Y.-I.L.; project administration, S.I.N. and Y.-I.L.; funding acquisition, S.I.N. and Y.-I.L. All authors have read and agreed to the published version of the manuscript.

**Funding:** This research was supported by Basic Science Research Program through the National Research Foundation of Korea (NRF) funded by the Ministry of Education (grant number: 2020R1I1A 1A01074184). The APC was funded by NRF (grant number: 2020R1I1A1A01074184).

**Conflicts of Interest:** The authors declare no conflict of interest.

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
