# Peer review of "Solution and Parameter Identification of a Fixed-Bed Reactor Model for Catalytic CO2 Methanation Using Physics-Informed Neural Networks"

_catalysts, doi:10.3390/catal11111304_

Round 1

Reviewer 1 Report

This work by Ngo and Lim presents a physics-informed neural network (PINN) mode to provide an isothermal fixed-bed model for catalytic CO2 methanation. This work is more on chemical engineering side. The overall quality of this manuscript is good, but the authors need to clarify in the manuscript the following aspects.

1) It's not very clear to me what exactly the input is for the two neural networks as shown in Figures 2 and 3. Both figures use "z" as the input. And the output from these two networks seems to be equivalent. Is this correct? From my understanding of the introduction, these two networks should have different input/output. 

It would be helpful to the readers if the authors can talk more about what the input/output looks like for these two networks. Otherwise, it seems very abstract for people not familiar with chemical engineering.

2) With the developed neural networks, one important question is how could people employ and benefit from them. It would be good if the authors can talk about the potential application scenario in section 5: summary and conclusions. 

Author Response

We thank the reviewers for their constructive and sincere comments. In this revised manuscript, we have addressed all the points raised by the reviewers. The main changes are as follows:

  1. 8 and Section 4.2 were added to explain the computational efficiency of the PINN.
  2. 9 – 11 in the revised manuscript were revised, restricting the region from 0 ≤ z ≤ 3 m to 0 ≤ z ≤ 2 m.
  3. According to the two reviewers’ comments, several sentences were revised or added.

Please find the response to the reviewer's question-by-question in the attachment. We also enclose the revised manuscript highlighted in blue for the texts changed by the authors.

Sincerely,

Reviewer 2 Report

Summary of work

This study implements a Physics Inspired Neural Network (PINN) to model ODEs governing CO2 methanation in an isothermal fixed bed reactor. The PINN was trained on data generated using a numerical solution of the ODEs. The trained PINN was then used to predict the solution of the ODEs at times outside the training region. The study further comments on the effect of the activation function, number of training set examples, neural network structure, and training time on the performance of the PINN. Finally, the trained PINN was used to back out the effectiveness factor by fitting to data generated in experiments. The experiments were represented by adding noise to the numerical solution of the ODEs.

Broad comments

The idea of applying PINNs to problems in chemical engineering is indeed exciting. However, given that this study only solves a trivial problem, whose solution can be easily obtained using a numerical solver, I cannot recommend the publication of this manuscript in its current form. This study could have been accepted if this was a first demonstration of this approach. But, as far more complex ODE/PDE systems have been solved using PINNs, as also noted by the authors (references 32, 34, and 35), I cannot recommend the publication of this work. I would be happy to recommend publication if the authors can demonstrate the application of PINNs to a more complicated problem relevant to catalysis/reaction engineering.

I would also recommend that the authors show the efficiency of this approach over solving the system numerically. This could be demonstrated by comparing the total compute time involved in training the PINN vs. solving the system of ODEs numerically. And I don’t think that using a PINN offers tremendous advantage in this simple case. And for this reason, I would like the authors to consider a more complicated example.

The study predicts the solutions only on a fixed set of parameters (T=450 oC, P= 5 bar, volumetric flow rate = 10 Nm3/s, and molar ratio of CO2 to H2=1/4). Earlier studies have used this approach to train a single PINN to predict the solution of ODEs for wide range of parameters. I believe this is where the power of PINNs over solving the ODEs numerically for every set of parameters lies. I would recommend that the authors include this in their revision.

Specific comments

  • Will it be possible to incorporate the boundary condition at z=0 without explicitly minimizing a 2nd loss function? One way of incorporating boundary conditions in PINNs that I am aware of, is to express the solution as Fi(z) = Fi(z=0) + z*(PINN(z)). I believe this is a more physical way to set up the PINN. Authors can comment on if there are issues with this implementation here.
  • Figures 2 and 3 are very similar and Fig. 3 can be moved to the SI.
  • As shown in Figure 8, most of the predictions made by the PINN are in the uninteresting region where the solutions have already reached the steady state. I don’t think this is an effective demonstration of the power of PINNs.

Author Response

(The authors gave the same response as above.)

Round 2

Reviewer 2 Report

Broad comments

The authors have addressed a few minor points in my review. However, they have not dealt with the major issues that I had highlighted. This work is a very small increment over existing work. I still cannot accept the paper in its current form and encourage the authors to pursue a more complicated system of ODEs. I will accept this paper if the authors analyze the same system with energy balances and demonstrate that a PINN trained on a few operation parameters can be applied to predict kinetics at other operating conditions. Below I comment on two major concerns that I would like addressed.

Novelty of this work

A coupled system of non-linear ODEs has also been solved in reference 29. In my opinion even that work is only a demonstration of the application of PINNs to chemical kinetics. It doesn’t solve any interesting problem. However, unlike this work, reference 29 is a first demonstration of PINNs in chemical kinetics and catalysis. And they outline a general framework to solve such problems. I do not think that this work is that great of an advancement, if any, over reference 29 to be warranted publication. Especially given that much more complicated systems have been solved. I would encourage the authors to implement an energy balance so that they can solve for these ODEs at different process conditions. The next step in this field, in my opinion, should be to demonstrate the application of PINNs to vastly more complicated systems rather than simple problems which can be trivially solved.

Efficiency of PINNs over numerically solving ODEs

In the revision, the authors did demonstrate that once trained, the PINN is more efficient than the numerical solution. But again, as the efficiency of PINNs over numerically solving ODEs has been demonstrated for far more complicated systems, this is not a new insight. Furthermore, the speed up is not useful for this problem given the extra time required to obtain data and train the neural network.

Author Response

We thank the reviewers for their constructive and sincere comments. In this revised manuscript, we have addressed major points raised by the reviewers. The main changes are as follows:

  1. Section 4.5 was added to explain the extension of PINN to various operating conditions.
  2. Conclusions were revised.

Please find the response to the reviewer's question-by-question. We also enclose the revised manuscript highlighted in blue (first revision) and green (second revision) for the texts changed by the authors in the attached file.
